# Probiotic *Bifidobacterium breve* MCC1274 Protects against Oxidative Stress and Neuronal Lipid Droplet Formation via PLIN4 Gene Regulation

**DOI:** 10.3390/microorganisms11030791

**Published:** 2023-03-20

**Authors:** François Bernier, Tatsuya Kuhara, Jinzhong Xiao

**Affiliations:** Next Generation Science Institute, R&D Division, Morinaga Milk Industry, Co., Ltd., Zama 252-8583, Japan; frankbernier@mac.com (F.B.);

**Keywords:** *Bifidobacterium breve*, PLIN4, lipid droplets, niacin, oxidative stress, inflammation

## Abstract

Consumption of *Bifidobacterium breve* MCC1274 has been shown to improve memory and prevent brain atrophy in populations with mild cognitive impairment (MCI). Preclinical in vivo studies using Alzheimer’s disease (AD) models indicate that this probiotic protects against brain inflammation. There is growing evidence that lipid droplets are associated with brain inflammation, and lipid-associated proteins called perilipins could play an important role in neurodegenerative diseases such as dementia. In this study, we found that *B. breve* MCC1274 cell extracts significantly decreased the expression of perilipin 4 (*PLIN4*), which encodes a lipid droplet docking protein whose expression is known to be increased during inflammation in SH-SY5Y cells. Niacin, an MCC1274 cell extract component, increased *PLIN4* expression by itself. Moreover, MCC1274 cell extracts and niacin blocked the PLIN4 induction caused by oxidative stress in SH-SY5Y cells, reduced lipid droplet formation, and prevented IL-6 cytokine production. These results offer a possible explanation for the effect of this strain on brain inflammation.

## 1. Introduction

Sporadic and familial Alzheimer’s diseases (ADs) are debilitating neurodegenerative conditions that involve the deposition of amyloid plaques and Tau tangles in the brain [1]. The tremendous efforts made by pharmaceutical companies to develop anti-amyloid therapies over the last 20 years have all failed to completely stop the disease progression [2], suggesting that our understanding of the function of amyloid beta and its role in AD etiology is far from clear and that novel strategies are needed to combat this disease. In recent years amyloid beta has gradually come to be recognized as an antimicrobial peptide [3] that is produced in response to brain inflammation and protects the brain against various pathogens. Brain inflammation also occurs in individuals with mild cognitive impairment (MCI), years before AD onset [4]. Increasing evidence suggests that brain inflammation is linked to gut microbiome dysbiosis. A leaky gut–blood barrier permits the release of gut microbes themselves or their toxins into the bloodstream, triggering the amyloid formation cascade and resulting in gradual neuronal cell death [5]. This sequence of events suggests that effective anti-inflammatory therapies could halt AD progression before it becomes irreversible [6].

Recent studies have shed light on how brain cell lipid metabolism undergoes profound changes in response to the oxidative stress experienced by cells during inflammation, as shown by lipid droplet (LD) accumulation in the brain cells of AD patients and in AD mouse models [7,8,9]. These LDs, which are rich in free fatty acids (FFAs), can serve as a transient energy source for neuronal cell mitochondrial beta-oxidation when energy is in high demand, such as during inflammation [10]. LD recruitment to mitochondria to serve as an energy source is mediated by a family of proteins called perilipins (PLIN1-5), which are expressed in various cells throughout the body [11,12]. Neurons and muscle cells rely on PLIN4 to enable mitochondria to use LDs for beta-oxidation under conditions of inflammation and oxidative stress [13]. Although beta-oxidation can generate more ATP than oxidative phosphorylation, it produces more peroxide. It also consumes more oxygen, potentially damaging neurons, which would explain why these cells preferentially use glycolysis and ketolysis to generate ATP under normal conditions [14]. It was recently reported that PLIN4 expression is induced when neurons are exposed to the toxin 1-methyl-4-phenyl-1,2,3,6-tetrahydropyridine (MPTP), which impairs mitophagy-mediated repair of damaged mitochondria; this finding supports the neuronal preference for energy production pathways that do not involve beta-oxidation [13].

Probiotics are live microorganisms that can potentially help to treat several mental illnesses [15]. They are “friendly” or “healthy” bacteria that we ingest through foods, beverages, and dietary supplements. Two of the most frequently reported benefits of probiotics are the anti-inflammatory effect that they exert on the central nervous system (CNS) through the gut–brain axis [16,17] and the correction of gut microbiota dysbiosis, which is related to CNS disorders, with poorly understood etiology, and infections [18]. 

Considerable effort has gone into studying the anti-inflammatory effects of the short-chain fatty acids (SCFAs) produced by probiotics (acetate, butyrate, and propionate) [19]. However, given that SCFAs exhibit low penetration of the brain, other important molecule(s) produced by probiotics are likely to be responsible for their apparent anti-inflammatory effects. Three decades ago, well before SCFAs were identified, certain probiotics such as *Lactobacillus* and *Bifidobacterium* were shown to produce and accumulate essential vitamins, such as niacin (vitamin B3) [20,21], that are linked to the regulation of inflammatory processes [22,23]. 

Recently, our group demonstrated that daily consumption of *Bifidobacterium breve* MCC1274 improved memory in individuals suffering from mild cognitive impairment (MCI), as well as preventing brain atrophy, in separate randomized double-blind placebo-controlled trials [24,25]. Furthermore, treating APPKI mice (AppNL-G-F) with the same probiotic reduced amyloid plaque deposition, increased alpha-secretase expression, and decreased microglial cell activation [26]. Plasma metabolomics analysis of these mice indicated that MCC1274 consumption increased the levels of several metabolites with known activity against oxidative stress, as well as metabolites related to the mitochondrial TCA cycle, although the specific metabolite(s) responsible for these effects were not identified [27]. In the present study, we asked whether other metabolites generated by this probiotic prevent the inflammation-derived oxidative stress and lipid droplet formation that occur in MCI and Alzheimer’s disease by controlling perilipin gene expression. 

## 2. Materials and Methods

### 2.1. Preparation of Live Bacterial Cells

*Bifidobacterium breve* MCC1274 cells (Morinaga Milk Research Center, Zama, Japan) were grown in MRS medium (Difco, Beckton Dickinson, Franklin Lakes, NJ, USA) for 16 h in an anaerobic chamber at 37 °C. The cells were then transferred to 50 mL Falcon tubes (Corning, Reynosa, Mexico) and centrifuged at 8000× *g* for 10 min. Next, the cell pellets were washed and centrifuged three times with PBS buffer to remove any traces of the MRS fermentation medium. The wet cell pellets were finally resuspended in PBS at 0.3 g per mL (equivalent to 1 × 10^8^ cells/mL).

### 2.2. Chemicals

Nicotinic acid (niacin, NA) was purchased from Tokyo Chemical Industry Co., Ltd. (Tokyo, Japan). Nicotinamide riboside chloride (NR) was purchased from eNovation Chemicals, LLC (Bridgewater, NJ, USA). Nicotinamide adenine dinucleotide (NAD+) and nicotinamide (NAM) were purchased from Merck KGaA (Darmstadt, Germany). Acetonitrile (high-performance liquid chromatography [HPLC] grade) was purchased from FUJIFILM Wako Pure Chemical Corporation (Osaka, Japan). Ammonium acetate (suitable for mass spectrometry) and 3-methyl-2-oxindole were purchased from Merck KGaA. Unless otherwise stated, all chemical reagents used were of analytical grade.

### 2.3. Preparation of Bacterial Cell Extracts

#### 2.3.1. Preparation of Heat-Killed Cell Extracts

MCC1274 cells suspended in PBS (1 × 10^8^ cells/mL) were heated in Eppendorf tubes on a heating block at 98 °C for 10 min; then, they were centrifuged at 4 °C for 10 min at 8000× *g*. The supernatant (referred to hereafter as “HKS”) was immediately stored at −80 °C for later use in the in vitro cell culture experiments.

#### 2.3.2. Preparation of Sonicated Cell Extracts

MCC1274 cells suspended in PBS (1 × 10^8^ cells/mL) were sonicated (one sonic pulse every 2 s) on ice in 15 mL conical polypropylene tubes (Corning, Reynosa, Mexico) for 1 h, and the sonicated products were then centrifuged at 4 °C for 10 min at 8000× *g*. The supernatant (referred to hereafter as “Sonic”) was immediately stored at −80 °C for later use in the in vitro cell culture experiments.

### 2.4. Cell Culture

Human neuroblastoma cells SH-SY5Y were obtained from ATCC, USA. The cells were cultured and maintained in Dulbecco’s Modified Eagle Medium (DMEM, Gibco, Grand Island, NY, USA), containing 4.5 g/L glucose, 4 mM glutamine, 10% heat-inactivated fetal bovine serum (Gibco Life Technologies, Grand Island, NY, USA), 25 mM hepes, and 0.1% (v/v) penicillin/streptomycin (Gibco Life Technologies) in an incubator with a 5% CO_2_ atmosphere at 37 °C. The cell medium was replaced every three days.

The bacterial supernatants (HKS or Sonic) were added to the confluent SH-SY5Y cells in 24-well plates to yield a final concentration of 1% v/v. The cells were then incubated for varying amounts of time in a CO_2_ incubator. Next, the cell medium was aspirated, 0.5 mL TRIzol (Ambion Life Tech., Austin, TX, USA) was added to cells, and the plates were incubated for 5 min at room temperature. The TRIzol–cell solution was then stored at −80 °C for later RNA extraction. For the experiments related to oxidative stress protection, H_2_O_2_ was added at the same time as the MCC1274 extracts or NA.

### 2.5. RNA Extraction 

Total RNA was extracted from the TRIzol-treated cells using a Qiagen RNeasy Plus Universal Mini kit (Qiagen, Hilden, Germany). After extraction, the RNA concentration and quality were determined using a Nanodrop One device (Thermo Scientific, Waltham, MA, USA).

### 2.6. PCR

cDNA was synthesized from 1 μg total RNA using a Takara PrimeScript Reagent kit (TakaraBio, Kusatsu, Shiga, Japan) per the manufacturer’s instruction. Real-time PCR was conducted using a BioRad PCR Thermal Cycler with a final reaction volume of 25 μL containing 50 ng cDNA, 1 μL of each primer pair (20 μM), and Takara TB Green Premix Ex Taq II (TII RNaseH Plus), as per the manufacturer’s kit protocol to determine the relative mRNA concentration for each gene of interest under each treatment condition; all reactions were performed in duplicate. The sequences of the primers used for the PCR reactions are shown in Appendix A. The relative RNA concentrations were determined using a standard curve generated using a preamplified PLIN4 PCR product or GAPDH. The PCR reaction conditions were as follows: 95 °C for 10 s, followed by 40 cycles of 95 °C for 20 s, 64 °C for 20 s, and 72 °C for 10 s.

### 2.7. Measurement of NA and Related Metabolites by Mass Spectrometry

The concentrations of the metabolites in the samples (HKS and Sonic) were analyzed using liquid chromatography–tandem mass spectrometry (LC–MS/MS; Vanquish HPLC connected with TSQ-FORTIS, Thermo Fisher Scientific, Waltham, MA, USA). Chromatographic separation was performed using an XBridge^®^ Phenyl column (Waters Corporation, Milford, MA, USA) (4.6 × 150 mm, 5 μm). Mobile phase A (containing 1 g/L ammonium acetate in water) and mobile phase B (methanol) were applied at a flow rate of 0.2 mL/min. Gradient elution was performed between 2% and 50% of phase B. Quantification was achieved by comparing the metabolite peak areas with the corresponding synthetic compound standards and an internal standard (3-methyl-2-oxindole). The precursor ion’s LC–MS/MS spectrum (product ion data) was evaluated to determine the final content of each metabolite (Table 1 and Appendix A).

### 2.8. FACS and LD Assay

To detect changes in the LDs following treatments, the cells were washed three times with 1X PBS then stained with BODIPY (2 μM) for 20 min. The cells were then trypsinized and collected for fixation with 4% formalin for 30 min, followed by resuspension in Bio-Rad staining buffer (Bio-Rad, Berkeley, CA, USA). We used a Becton Coulter FACS analyzer to detect the average BODIPY signal intensity (FITC) of 10,000 individual cells for each treatment condition (n = 4). This measurement indicates the number of LDs present in each cell. The siRNA used to knock down human PLIN4 mRNA expression was purchased from Merck Millipore and transfected into cells using Lipofectamine RNAiMAx reagent (ThermoFisher, Waltham, MA, USA), as per the manufacturer’s instructions.

### 2.9. Microscopy

Neuroblastoma SH-SY5Y cells were plated at a density of 150,000 cells/well in a microscopy chamber slide and incubated overnight. The next day, compounds and bacterial extracts were added to cells in the presence of H_2_O_2_ (200 μM), and the cells were cultured for another 24 h. The following day, the cell medium was aspirated, and the cells were washed three times with PBS before adding BODIPY staining solution (2 μM) to visualize the lipid droplets. After 20 min of staining, the cells were again washed three times with PBS, fixed for 30 min in 4% paraformaldehyde, and then mounted with Hard Set mounting media containing DAPI (Vector Laboratories, Burlingame, CA, USA) for microscopic analysis (Olympus BX 53).

### 2.10. Statistical Analysis

For the in vitro cell experiments, statistical analysis was conducted using Microsoft Excel. Significant differences between treatment conditions and controls were determined using the Student’s t-test for two-group comparison and non-repeated measures ANOVA with the Bonferroni post hoc test for multiple-group comparison, respectively.

## 3. Results

### 3.1. Metabolites in B. breve MCC1274 Extracts Specifically Reduce PLIN4 Expression in Human Neuroblastoma Cells

To test whether *B. breve* MCC1274-derived metabolites affected perilipin gene expression, SH-SY5Y human neuroblastoma cells were exposed to the two cell extracts (HKs and Sonic) at final concentration of 1% v/v for 1 h, total RNA was extracted, and RT-PCR was conducted to analyze the changes in perilipin mRNA expression. As shown in Figure 1A–E, the metabolite(s) contained in both extracts (HKS, Sonic) reduced PLIN4 expression but not the expression of any other perilipin at the mRNA level. The HKS preparation method, due to its simplicity, was chosen for further experiments.

### 3.2. NA and B. breve MCC1274 Extracts Reduce PLIN4 mRNA Expression in SH-SY5Y Neuroblastoma Cells

A recent study suggested that the tryptophan metabolite NA may reduce lipid droplets in microglia cells [28]. Given that certain *Bifidobacterium* strains can produce NA [21], we performed mass spectrometry of *B. breve* MCC1274 extracts to detect NA and its metabolites nicotinamide (NAM), nicotinamide riboside (NAR), and nicotinamide adenine dinucleotide (NAD). As shown in Table 1, both the HKS and Sonic preparations contained around 1 µg/mL of NA, which indicates that the 1% (v/v) extract concentration that we used for the SH-SY5Y cell treatment experiments was equivalent to 100 nM of NA.

While the *B. breve* MCC1274 extracts contained sufficient NA to reduce PLIN4 expression, they might also contain substances that generally do not reach the brain and could negate the effect of the NA. Therefore, we tested whether the purified NA had the same effect as the cell extracts by exposing the SH-SY5Y cells to 1% (v/v) *B. breve* MCC1274 extract, NA, or NAM for 1 h. As shown in Figure 2, *B. breve* MCC1274 HKS (1%) and NA (30 nM and 100 nM) caused a rapid decrease in the PLIN4 mRNA expression after 1 h of exposure. While treatment with NAM also decreased the PLIN4 mRNA expression, the effect did not reach significance. Treatment with NAR or NAD did not reduce the PLIN4 mRNA expression.

### 3.3. NA and B. breve MCC1274 Extracts Protect against Oxidative Stress Caused by Exposure to H_2_O_2_

Brain cell damage in neurodegenerative diseases is linked to increased oxidative stress and LD formation [29]. Therefore, we hypothesized that NA and *B. breve* MCC1274 extracts might protect SH-SY5Y cells from exposure to peroxide (H_2_O_2_), a very potent inducer of oxidative stress, since they both reduced the mRNA expression of *PLIN4*, which encodes a lipid droplet chaperone protein. As shown in Figure 3, exposing the cells to H_2_O_2_ for 24 h caused an increase in the PLIN4 expression that was blocked by treatment with either the NA or *B. breve* MCC1274 extracts.

### 3.4. NA and B. breve MCC1274 Extracts Reduce LD Formation after Exposure to H_2_O_2_

We then looked at changes in the lipid droplet formation when neuroblastoma cells were exposed to H_2_O_2_ using fluorescence-activated cell sorting (FACS). We first confirmed that 200 µM H_2_O_2_ increased the LD formation in SH-SY5Y cells [30] using BODIPY staining followed by FACS analysis. We then exposed the cells to NA or *B. breve* MCC1274 extracts (HKS) with 200 µM H_2_O_2_ for 24 h, using a random siRNA as a control and PLIN4 siRNA as a positive control. Oleic acid was also used as a control to induce LD formation (Figure 4, Appendix A). The data indicated that the NA produced by *B. breve* MCC1274 suppressed the LD formation caused by oxidative stress, and this effect was dependent on a selective reduction in the PLIN4 mRNA expression.

### 3.5. NA and B. breve MCC1274 Extracts Prevent the Induction of IL-6 Expression

When brain cells are exposed to oxidative stress, they not only exhibit increased LD production but also, as reported previously [13], release inflammatory cytokines such as IL-6 as a distress signal directed toward microglia [31]. Using the same H_2_O_2_ exposure conditions, we found that NA and MCC1274 HKS extracts blocked the IL-6 cytokine production induced by oxidative stress, suggesting that NA and *B. breve* MCC1274 prevent H_2_O_2_-induced cell damage (Figure 5).

## 4. Discussion

Canadian pathologist Rudolf Altschul first identified NA more than 65 years ago as having lipid-lowering properties [32]. NA is still the focus of intensive research as researchers try to understand its numerous positive effects on the human body, particularly the brain and the immune system. Many studies have demonstrated the potential ability of NA to modulate inflammation [22,23,33]. Earlier this year, Moutinho et al. showed that the induction of HCAR2 expression by NA modulated the microglial response and limited disease progression in a mouse model of Alzheimer’s disease [28]. When the NA receptor expression is eliminated, amyloid beta accumulates in mouse brains, indicating a positive role for this receptor in the stress response. When these mice were treated with NA, robust clearance of amyloid plaques was observed, suggesting that NA may repolarize microglial cells from an M1 (proinflammatory) to an M2 (noninflammatory, phagocytic) type. Our findings that NA blocked oxidative stress, LD formation, and IL-6 expression in neuroblastoma cells were consistent with this earlier finding. The results from our previous study of APP KI treated with *B. breve* MCC1274 also align with this observation that NA can directly affect brain cells experiencing oxidative stress. In our mouse study, *B. breve* MCC1274 blocked activation of IBA1 expression by microglial cells. Mice that consumed *B. breve* MCC1274 also had a lower amyloid plaque burden than the control mice and performed better in a memory-related behavioral test [26]. Blood metabolomics identified multiple molecules such as soy isoflavones (e.g., genistein), indole derivatives of tryptophan (e.g., 5-methoxyindoleacetic acid), and other metabolites with potent antioxidative activities in the group of mice that received probiotics. In addition, the levels of glutathione-related metabolites and TCA cycle–related metabolites that could decrease brain oxidative stress were also elevated. Unfortunately, NA could not be detected in the blood due to its short half-life [34].

A clinical trial is currently underway to determine the potential of NA to treat Alzheimer’s patients (https://clinicaltrials.gov/ct2/show/NCT03061474 (accessed on 3 December 2022)). Our group recently conducted two double-blind placebo-controlled trials in individuals suffering from MCI and showed that *B. breve* MCC1274 could ameliorate memory and prevent brain atrophy [24,25]. The enhanced cognitive function and reduced brain atrophy that we observed in individuals who consumed *B. breve* MCC1274 daily might be associated with NA production. Interestingly, Chellappa et al. reported that the gut microbiome converts host-derived nicotinamide into nicotinic acid, maintaining circulating nicotinic acid levels even in the absence of dietary consumption of nicotinamide [35]. This new finding implies that consumption of *B. breve* MCC1274 could enhance the host’s ability to produce NA and induce a reduction in PLIN4 expression even in the absence of oxidative stress conditions, as we observed (Figure 1D).

Our finding that NA production by *B. breve* MCC1274 directly impacts LD formation by reducing PLIN4 expression reduction is interesting given that LD accumulation is a common phenomenon in tumor cells [36]. It also has come to light recently that a higher intratumoral level of PLIN4 is associated with lower survival rates in patients with colorectal or endometrial cancer (https://www.proteinatlas.org/ENSG00000167676-PLIN4/pathology (accessed on 3 December 2022)). Interestingly, these two organs are located close to the intestine, which contains a microbiota known to harbor *Bifidobacterium* [37], and a lower abundance of *Bifidobacterium* near colorectal and endometrial tissues has been linked to a higher incidence of these two cancers [38,39]. We think there is a possibility that NA derived from the *Bifidobacterium*-rich endometrial and colon microbiotas may protect against these cancers by regulating LD formation by cancer cells.

Our study had some limitations that we plan to address soon. First, we would like to understand whether the effect of the MCC1274-derived NA on the brain is mediated by the vagal nerve or via a more classic blood absorption route. Second, we would like to determine whether *B. breve* MCC1274 can restore the impairment of mitochondrial mitophagy associated with increased PLIN4 expression [13]. Moreover, we would like to investigate whether consuming *B. breve* MCC1274 impacts mitophagy [40,41,42] related to long-term potentiation (LTP) [43] during brain inflammation and under normal conditions in vivo.

## 5. Conclusions

In conclusion, our results demonstrate a novel regulatory mechanism(s) by which *B. breve* MCC1274 reduces PLIN4 expression and LD formation in neuroblastoma cells undergoing oxidative stress. We also showed that the molecule produced by *B. breve* MCC1274 that is responsible for blocking LD formation and the resulting inflammation is NA. Therefore, daily consumption of *B. breve* MCC1274 may protect against brain inflammation in people suffering from MCI and dementia, such as those with Alzheimer’s disease.

## Figures and Tables

**Figure 1 microorganisms-11-00791-f001:**
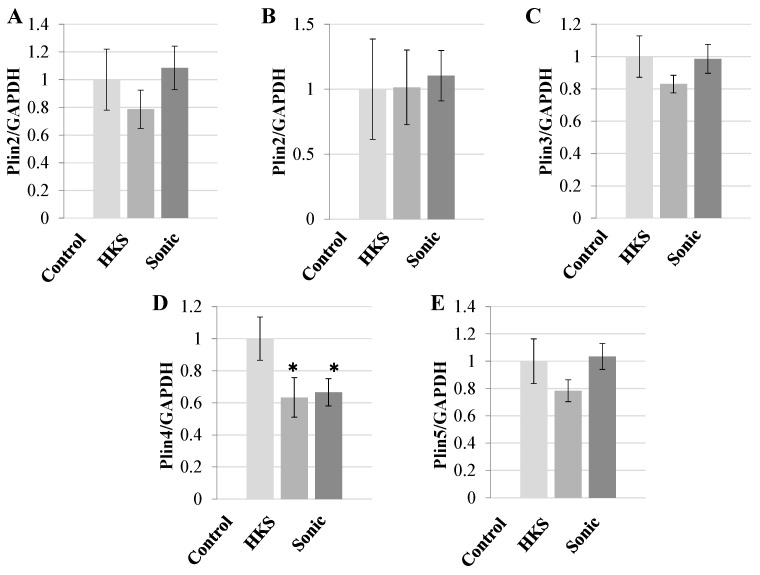
The effects of *Bifidobacterium breve* MCC1274 extract on perilipin gene expression in SH-SY5Y cells. The SH-SY5Y cells were exposed to *B. breve* MCC1274 heat-killed supernatant (HKS) or sonicated supernatant (Sonic) for 1 h at a 1% v/v final concentration. The expression of human perilipin mRNA genes relative to human GAPDH by RT-PCR (**A**–**E**). The data are shown as the mean +/− standard deviation (SD) of six replicates. * *p* < 0.05, vs. Control (non-repeated measures ANOVA with the Bonferroni post hoc test).

**Figure 2 microorganisms-11-00791-f002:**
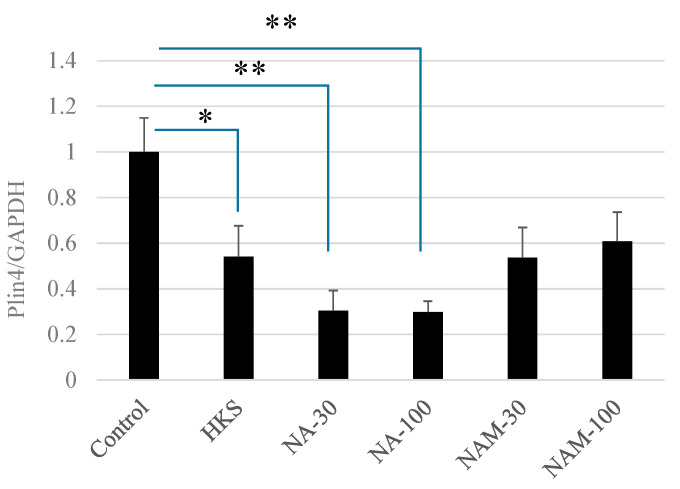
*Bifidobacterium breve* MCC1274 extracts and niacin reduce PLIN4 mRNA expression in SH-SY5Y neuroblastoma cells. The SH-SY5Y cells were exposed to HKS (*B. breve* MCC1274 heat-killed cell extracts, 1% final v/v), niacin (NA), or niacinamide (NAM) (final concentration of 30 or 100 nM) for 1 h. Human PLIN4 mRNA expression relative to human GAPDH was determined by RT-PCR. The data are shown as the mean + SD of three replicates. * *p* < 0.05, ** *p* < 0.01 (non-repeated measures ANOVA with the Bonferroni post hoc test).

**Figure 3 microorganisms-11-00791-f003:**
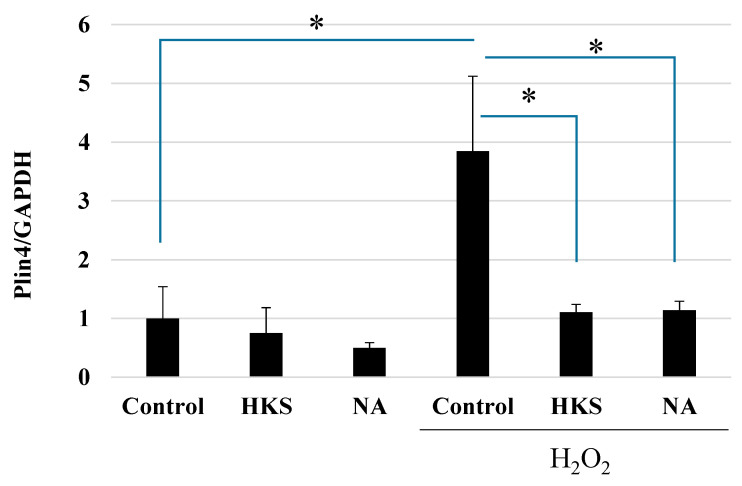
*Bifidobacterium breve* MCC1274 extracts and niacin protect SH-SY5Y cells against oxidative stress caused by exposure to H_2_O_2_. The SH-SY5Y cells were exposed to HKS (*B. breve* MCC1274 heat-killed cell extracts, 1% v/v) or niacin (NA, 100 nM) with or without H_2_O_2_ (200 µM) for 24 h. Human PLIN4 mRNA expression relative to human GAPDH expression was determined by RT-PCR. The data are shown as the mean + SD of four replicates. * *p* < 0.05 (Student’s t-test for two-group or non-repeated measures ANOVA with the Bonferroni post hoc test for multiple-group comparison, respectively).

**Figure 4 microorganisms-11-00791-f004:**
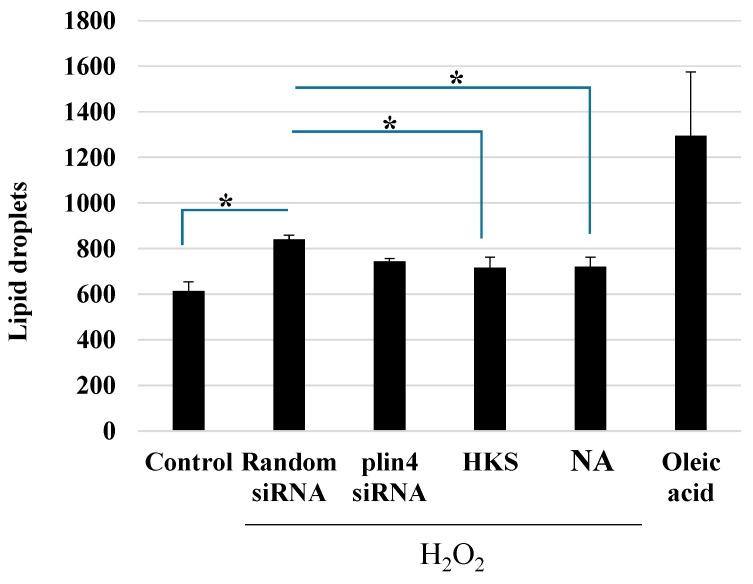
*Bifidobacterium breve* MCC1274 extracts and niacin suppress lipid droplet formation in SH-SY5Y cells. The SH-SY5Y cells were exposed to siRNA (random siRNA or PLIN4 siRNA), HKS (*B. breve* MCC1274 heat-killed cell extracts, 1% final v/v), or NA (100 nM) in the presence of H_2_O_2_ (200 µM) for 24 h and then stained with BODIPY (2 µM). Untreated cells were used as a control. Oleic acid (60 µM) was used as a control to confirm the induction of lipid droplets. The *y*-axis shows the FITC signal as measured by flow cytometry. The data are shown as the mean + SD of three replicates. * *p* < 0.05 (Student’s t-test for two-group or non-repeated measures ANOVA with the Bonferroni post hoc test for multiple-group comparison, respectively).

**Figure 5 microorganisms-11-00791-f005:**
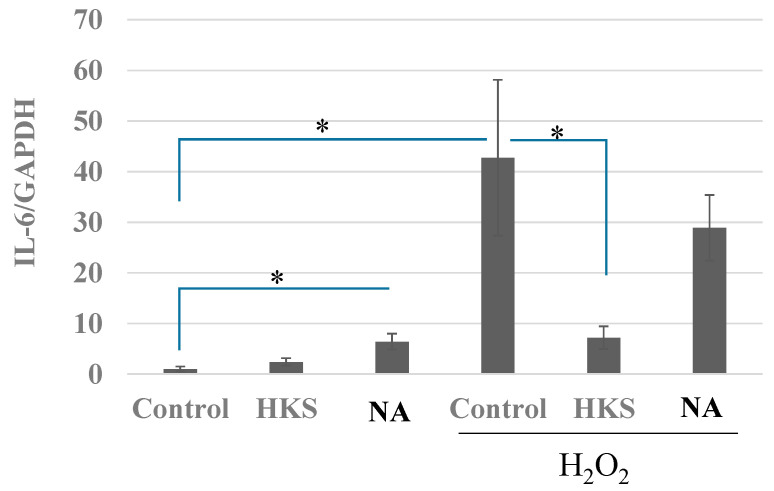
*Bifidobacterium breve* MCC1274 extracts and niacin prevent the increase in IL-6 expression induced by oxidative stress. The SH-SY5Y cells were exposed to HKS (*B. breve* MCC1274 heat-killed cell extracts at 1% v/v final concentration) or niacin (NA, 100 nM), with or without H_2_O_2_ (200 µM), for 24 h. Human interleukin-6 (IL-6) mRNA expression relative to human GAPDH expression was determined by RT-PCR. The data are shown as the mean + SD of four replicates. * *p* < 0.05 (Student’s t-test for two-group or non-repeated measures ANOVA with the Bonferroni post hoc test for multiple-group comparison, respectively).

**Table 1 microorganisms-11-00791-t001:** Concentration of Niacin-related metabolites in cell preparation of Bifidobacterium breve MCC127.

	HKS	Sonic
NA	1.43	1.12
NAM	7.01	14.8
NR	0.25	0.2
NAD+	53.35	23.68

Metabolite abbreviations are as follows: NAM (niacinamide or nicotinamide), NA (niacin or nicotinic acid), NR (niacinamide riboside) and NAD+ (nicotinamide adenine dinucleotide). Sonic (sonicated preparation) and HKS (Heat-treatment preparation). Concentrations are shown in µg/mL.

## Data Availability

Data available on request.

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
