# Peer review of "Probiotic Bifidobacterium breve MCC1274 Protects against Oxidative Stress and Neuronal Lipid Droplet Formation via PLIN4 Gene Regulation"

_microorganisms, 2023, doi:10.3390/microorganisms11030791_

Round 1

Reviewer 1 Report

Bernier reported that Bifidobacterium breve MCC1274 protects against oxidative stress and neuronal lipid droplet formation through decreasing gene expression of PLIN4. Cell extracts or niacin blocked PLIN4 induction caused by oxidative stress in SH-5YSY cells and blocked the formation of lipid droplets. These processes are described as the anti-inflammatory mechanisms in the brain tissue of this strain. The study of this paper is logically structured, and the results support the authors' hypothesis. The several points, however, need to be clarified. In particular, my main question is whether the niacin levels of MCC1274 are unique in compared with other bifibacterium. The authors may have been in a hasty preparation of the manuscript, and there are several mistakes throughout, requiring another careful check of the manuscript.

1. Line 176: Please state your rationale for using two different types of treated cells, HKS and Sonic. All graphs in Fig. 1 are missing units on the vertical axis. In addition, more detailed conditions should be provided for sonication treatment in the materials and methods.

2. Line 102 and 181: The authors indicate the following "1% of the two separate extracts were then exposed for 1 hour". I don't understand what the one percentage means, relative to the number of bacteria or to the volume of the media?

3. Line 203: The concentrations of niacin in HKS and Sonic seem to be similar levels. If the antioxidant properties of niacin are a direct mechanism of HKS-MCC1274 strain, how can you interpret the results in Fig. 1D? Please describe exactly in the discussion part.

4. Lines 299-303: The author mentioned that consumption of MCC1274 improves the capacity for Niacin production in the host, but then the benefits of Niacin produced by MCC1274 itself, which you showed in this paper, are meaningless? Please be consistent in your discussion within the paper and be clear in your arguments.

5. In addition, could you explain how changes in niacin levels in the gut affect inflammation in the distal part of the brain? I think the limitation and future experiments in this paper is to be more specific on this point than mitochondrial mitophagy etc.

Minor comments

1.       Line 83: "g" is missing throughout the manuscript.

2.       Line117: 1 x10^8 cells/ml.

3.     Table 1: Horizontal lines in the table should also be drawn under “HKS and Sonic”. The units (µg/mL) should be listed in the table. Please revise as follows: “Concentration of Niacin-related … MCC 1274”.

4.       Line 226: H2O2; “2” should be subscripted.

5.  The statistical analysis of Figure 2-5 is not suitable and should be reanalyzed by a multiple comparison test, such as Dunnett's test.

6.      Please unify the labels of Niacin or NA in each graph. The color of the text in each graph as well.

Author Response

We are grateful for the reviewers for their thorough and thoughtful comments, which have proven invaluable in allowing us to strengthen our manuscript. We have given our best effort and believe that we have addressed all concerns of the reviewers. Our response to each of the referee’s comments are provided per attached file.

Reviewer 2 Report

This article found that probiotic Bifidobacterium breve MCC1274 and its cell extracts (containing Niacin) protects against oxidative stress and lipid droplet formation via PLIN4 gene regulation in human neuroblastoma SH‐SY5Y cells. However, the authors need to address the following points:

1.      The beneficial effects of probiotics are known to be strain-specific. I wonder if the modulatory effects described in this study are specific to MCC1274? Authors should include strains belonging to B. breve as well as other Bifidobacterium spp. as comparison groups.

2.      Fig. 1: Does the MCC1274 cell extract reduce PLIN4 mRNA in a dose-dependent manner?

3.      Please provide more details about the protocol of the H2O2 study. Is H2O2 used as a pre-treatment or is it co-treated with HKS and NA?

4.      Is there any beneficial difference between HKS and NA? As shown in Figure 5, HKS reduced the expression of IL-6 to a greater extent than the NA group. Moreover, under the absence of H2O2 conditions, NA can significantly induce the expression of IL-6 rather than HKS.

5.      What is the survival rate of SH-SY5Y cells during the study? The toxicity of HKS and NA are needed to address.

6.      In this study, more than two groups for statistical analysis, one-way analysis of variance should be used rather than Student’s T-test.

7.      The format, style and label of the figures should be consistent, please correct. There are many typos (e.g. SH-5YSY and 5Y-SHSY) in the manuscript. I strongly recommend that the authors send this manuscript to an expert in English editing and academic writing.

8.      Table 1 caption: niacin-related; Bifidobacterium; MCC1274

Author Response

(The authors gave the same response as above.)

Reviewer 3 Report

This paper is an important contribution and I recommend that it be accepted for publication.

Author Response

(The authors gave the same response as above.)

Round 2

Reviewer 2 Report

Minor comments:

Please confirm again that the name of human neuroblastoma cell mentioned in the manuscript is SH-SY5Y.

Author Response

We would like to thank you for your kind review of our manuscript. The manuscript was reviewed by a specialist.